# Creating a Supportive Work Environment: A Cognitive Behavioral Approach for Nurse Leaders

**DOI:** 10.3390/nursrep15030091

**Published:** 2025-03-06

**Authors:** Nurit Zusman, Caryn Scheinberg Andrews, Vladislav Kaslin, Anna C. Kienski Woloski Wruble

**Affiliations:** 1Henrietta Szold School of Nursing of Hadassah, Hebrew University, Jerusalem 9112001, Israel; candrews613@gmail.com (C.S.A.); annawruble@gmail.com (A.C.K.W.W.); 2The Galilee Medical Center, Nahariya 22000, Israel; vladislavkaslin@gmail.com

**Keywords:** cognitive behavioral techniques, management, transformational leadership, nursing, challenges, thought, coping

## Abstract

Purpose: This article focuses on identifying cognitive behavioral (CB) techniques that can help nurse supervisors more effectively navigate interpersonal challenges, reduce workplace stress, improve team cohesion, and, ultimately, enhance overall organizational performance and staff well-being. Approach: Through a comprehensive review of leadership literature and clinical management practices, we determined that CB techniques could be integrated into nursing management. Two hypothetical scenarios within this context are offered, where CB techniques can enhance nursing leadership effectiveness. Conclusions and Recommendations: CB techniques offer a humanistic approach to nursing leadership through: (1) providing tools for leaders to reframe challenges and frustrations, particularly in resource-limited settings; (2) offering stress-management strategies for nursing leaders; and (3) enhancing communication skills, self-awareness, and team motivation. These applications can potentially improve both staff and management satisfaction, ultimately improving patient care quality. Healthcare organizations should consider incorporating CB techniques into their leadership development programs. We suggest practical ways to implement these techniques in daily nursing management, emphasizing the importance of creating supportive and safe work environments and provide recommendations for future research. This perspective extends the cognitive behavioral approach beyond its traditional therapeutic context into nursing leadership, providing a novel theoretical framework for understanding and enhancing leadership development in healthcare settings.

## 1. Introduction

The American Nursing Association (ANA) promotes the concept of transformational leadership for clinical and administrative nursing management [1]. Transformational leadership [2] revolves around the attributes and behaviors that management need in order to motivate and empower team members. Transformational leadership requires ongoing role-specific support to aid supervisors to fulfill their roles in both everyday and crisis situations [3]. It has been reported that nurse supervisors lack adequate skills to meet transformational roles. They struggle to fulfill transformative roles due to a lack of adequate training and preparation for the role, an absence of tailored leadership development programs, and, at times, a toxic work environment including bullying and poor relationships and a lack of systemic support for their mental and physical health [4]. Therefore, nursing management needs new and creative leadership skills to develop and promote their nursing teams, which may ultimately have the potential to enhance nurse and patient outcomes [5,6]. Transformational leadership supports empowering and promoting nurse engagement. By utilizing authentic and transformational leadership, and protecting the health of the workforce, competent leaders can reduce nurse burnout [7].

Transformational leadership, described by the sociologist [8] and further described by [9], extrapolates upon four I’s: *Idealized Influence*, whereby the leaders themselves model the behavior that is desired; *Individual Consideration*, or the leader’s ability to show concern and compassion; *Inspirational Motivation*, or creating a clear goal using simple language and inspiration, and *Intellectual Stimulation*, whereby the leader creates a culture of creativity and problem-solving skills to promote an environment where people work together cohesively. Although transformational leaders may work, at times, with individuals, the nature of leadership has an overall effect on the environment itself [10].

Transformational leadership involves integrating different styles of leadership and competencies. Transformational leadership is characterized by enthusiasm, commitment, and creativity [11]. A transformational leader must possess emotional intelligence, communication skills, collaboration skills, coaching skills, and mentoring skills. Supervisors can be helped by training in and implementing CB techniques.

In an organic interface with the cognitive behavioral approach, *Idealized Influence* can be described in the aspect of CB techniques whereby the leader or nurse supervisor models the desired behavior. Regarding *Individual Consideration*, the leader or supervisor is encouraged to use the techniques of “awareness of personal values” and “self-compassion”. In *Inspirational Motivation*, the leader uses the technique of normalization, or “end method”, to promote motivation. *Intellectual Stimulation* can be operationalized using a CB technique whereby the leader attempts to increase team cohesion effort and promote working toward a unified goal.

Challenges for nursing management include workplace stress, personnel conflicts, lack of time, and poor support [12]. All of these demands have been shown to contribute to the development of burnout among nurses and nursing supervisors [13,14]. Nursing supervisors have realized that focusing on issues such as poor staffing, burnout, and nursing shortages does not necessarily lead to a better workplace milieu [15]. In an organizational setting, nursing supervisors are compelled to use several strategies to reduce stress and improve work and performance and the environment [16,17].

With the goal of presenting a transformational leadership style, using workplace cases, we explore several challenging managerial situations to illustrate the application of CB communication techniques in improving the situation, both from the supervisor’s perspective and that of the nurse employee. The aim of this paper is to offer unique suggestions for nurse supervisors, focusing on the cognitive behavioral approach, which gives an interventional framework for supervisors, promoting a work environment with what is known as ‘soft skills’, that encourage open communication, collaboration, and active management of challenges.

## 2. What Are CB Techniques?

Cognitive behavioral techniques are based on promoting interaction and communication between nursing teams and management, and interpersonal relationships between nurses within the team, in order to foster interaction and communication. The goal is for these relationships to be strong and collaborative, leading to successful goal achievement [18]. CB techniques are used with individuals or groups with diverse levels of education and income, and are suitable for all ages, in various roles, and in diverse cultures [19]. In addition to primary care practices and other health frameworks, CB techniques have been used in various settings such as schools, vocational programs, and prisons [20,21,22].

CB techniques can be used to address various workplace problems and symptoms, at different levels of complexity. CB techniques employ the ABC Model, which includes an activating event (A), personal beliefs (B), and the consequences (C) of the person’s behavior [23], to evaluate automatic thoughts, irrational beliefs, or cognitive distortions that can lead to errors [24]. Through a slow process, automatic thoughts can be tested, refined, and modified [25]. Ellis [26] developed the concept that irrational beliefs or maladaptive/useless thoughts are based on an activating event that may lead to negative consequences [25].

Activating events are triggers, or a person’s interpretation or inference of that event [27]. People, places, things, situations, and implicit memories are all potential activating events [28]. Beliefs, core beliefs, or central assumptions or schemas are specific types of personal views [29] that represent the world, the future, and the self [30,31]. Core beliefs are developed early in life, are difficult to change, and are reflected in automatic thoughts [30]. When a belief is held, it has both emotional and behavioral consequences [27]. CB techniques are applied as effective strategies for regulating emotions. Strategies include “reappraisal and acceptance”, where it is possible to regulate emotions by choosing the situations in which “to be” or “not be”. Based on the emotional outcomes that are expected, situations are modified as they arise, with attention paid to specific aspects of those situations. Appraisal is then altered—bringing on a change in physiological, experiential, and behavioral responses [32,33].

Another CB technique that can be helpful for supervisors is the self-expansive mode [25], which helps in developing the ability to feel better about oneself, increasing self-esteem. The main goal of the self-expansive mode is to convert negative self-beliefs into positive evaluations of the self. For example, automatic thoughts, reactions, and beliefs, such as “I must do” or “I have to work harder, otherwise, I’m worthless” are transformed into more positive self-assessments [25,34]. Through this technique [29], the transformational leader can operationalized the concept called *Individual Consideration.*

## 3. CB Techniques and Nursing

Research supports the use of CB techniques that are applied in leadership practices within real-world nursing settings in several ways: by nurses with patients [14], by faculty as a strategy for teaching [35], and for students in their educational process [36]. There is evidence that leadership behaviors have a positive impact on various aspects of leadership, with positive implications for the organization [37,38,39], but there is no evidence of this in nursing.

Table 1 presents a summary of CB techniques and their expected outcomes that can be suggested for nursing supervisors. The following section offers two cases. The cases are examples of nursing conflicts that can be found in hospital departments, community settings, etc. Conflicts may arise between nurses, between patients/families and nurses, between nurses and other healthcare professionals, between supervisors and staff nurses, or between nurses and hospital administrators. Conflict-resolution strategies based in CB techniques, the goals of this manuscript, are offered as possible strategies to handle these conflicts and ensure a safe working environment for nurses. An analysis of possible automatic thoughts and emotions, behaviors, and consequences, and suggestions of how CB techniques could be individualized and implemented for each case is offered.

## 4. Case I: Management Within the Department

A life-threatening nursing error was discovered in the internal medicine department by the nurse supervisor. After a few days, Nurse T, who had started working at the department 6 months previously, was discovered to have made the error, and was summoned to a disciplinary hearing. Upon her return to the ward, she informed her colleagues that her license was being threatened to be revoked. The colleagues also found out that Nurse T had a history of medical errors. Colleagues of Nurse T had various reactions.

### 4.1. Nurse A

*Thoughts of nurse A*: “Why should I even try?” “The less I do, the lower my chances of making an error”. *Emotions of Nurse A*: Fear, anxiety, lack of self-confidence, fear of loss of respect of staff members. *Behavior of Nurse A*: Avoiding activities that require taking responsibility, for example, taking a stand when disagreeing with a medical order.

### 4.2. Nurse B

*Thoughts of Nurse B*: “This department is dangerous”. “Why didn’t they do something with Nurse T a long time ago?” “You don’t care, you only care about yourself”. *Emotions of Nurse B*: Anger, rage, fear. *Behavior of Nurse B*: Aggression, tantrums, preoccupied with anger rather than work.

*Consequences*: These dynamics can cause serious damage to departmental functioning. For example: One nurse went for medical treatment in the emergency room due to chest pain. Two nurses announced that they would not come to work the next day due to illness.

### 4.3. Case Analysis

*Nurse A*: Hospitals and clinics, though different work environments, are intense and stressful [40]. As can be seen from this case, the stress can stem from events that happen to other people in the department, not necessarily from an event that happened to the nurse herself. As such, these stressors can *trigger nurses to take actions that can lead to conflicts*. There are nurses who take responsibility, striving to deliver the finest care possible, and yet, are affected by irrational thoughts. Nurse A’s thoughts may indicate other thoughts such as: “I made some errors in the past and I should get punished too”. “I could have prevented this. I saw her, as a new nurse. She did things with excessive confidence, and even when the other nurses wanted to show her and help her, she refused. I should have done something. I feel guilty”. “ I’m not worthy to be a nurse. I should get punished too”. “Paying the price” is part of her own thought process. Even though she did not make the error, she incorporates it into self as if she did, based on her core beliefs—“I’m guilty [too]”.

*Nurse B*: Some nurses may demand that less experienced nurses learn from their own errors and learn from a senior nurse’s experiences, as seen in the scenario with Nurse B. Nurse B’s thoughts are “The problem is that I always thought the nurse who made an error was dangerous. The system and the department are dangerous, and no one cares. The nurse could have killed half the patients, and no one would care”. These thoughts can trigger behavior that is aggressive and angry. Even though Nurse B did not make the error, she incorporates into herself irrational thoughts that the ward is unsafe, and, therefore, she should not work there. Nurse B’s core beliefs for the world are—“The world is a dangerous place”.

### 4.4. CB Techniques for Case 1

Through open dialogue, listening for and identifying irrational beliefs without judgement, a *positive vision* can be formulated, instead of a negative one, reflecting the *Individual Consideration* applied in transformational leadership. One CB technique, called “negative filter”, reflects a person’s perception of a situation only through “bad–negative glasses”, without seeing the good. Only perceiving the future as negative in a particular situation can be interpreted as an irrational thought since there will not necessarily be a similar negative situation in the future [41].

The nurse supervisor, from her previous work with Nurse A, understands Nurse A’s automatic thoughts, and the subsequent irrational beliefs concerning this event. Nurse A is acting in a manner of avoidance and considering not coming to work; however, the supervisor can help her to see that her thoughts are a form of “black and white thinking”, resulting in her self-perception that she is a “bad” nurse. The supervisor may consider not just offering platitudes to mitigate the situation quickly, but rather, using listening skills and questions to evoke growth, and the *Inspirational Motivation* of transformational leadership. For example, she could ask: “Do you really think you are a bad nurse? Why?” In this way she helps build the nurse’s confidence in herself and improves the nurse’s work ethic and personhood. The supervisor employs a known communication skill called “mirroring” in order to encourage self-reflection on behaviors with the potential to make change if so desired.

For Nurse B, although the same preliminary techniques to understand her are employed, based on Nurse B’s core beliefs that the world is a dangerous place, the nurse supervisor could use a different approach—the CB technique called *normalization*, to encourage Nurse B to understand that it is normal to feel bad when a colleague makes a mistake. The nurse supervisor, employing the *Individual Consideration* of transformational leadership, helps Nurse B to cope by validating that her feelings are normal and expected, but reacting aggressively and with anger will not be beneficial. She helps her with insight to find less aggressive behavior to help mitigate the emotions, which came from irrational thoughts and caused the aggression and anger. Using “mirroring” once again, the nurse supervisor can help Nurse B to understand that she has choices in her expression of feelings; that her emotions of anxiety and anger are valid but that she may want to express them with words using a different tone or using examples to explain her feelings. Once again, the use of inquiry as a technique may encourage reflection and change.

Moreover, there is a cumulative effect on the department based on the possibility that Nurse A and Nurse B are suffering personally from stress and burnout, as well as anxiety or depression [42]. This combination of individual beliefs has an effect on the overall functioning of the department and is a challenge for the nurse supervisor. The change brought about by the use of CB techniques and a deep understanding of transformational leadership has the potential to make the nurses on the unit feel empowered, creating a better work environment.

## 5. Case II: Decision-Making for Senior Nursing Management

A nursing supervisor announced that there was a plan to transfer a few nurses from one internal medicine department to another internal medicine department. This change was being dictated by administrative pressure on the nurse supervisor. Nurses C and D are afraid of being transferred from their current position, but express it differently.

### 5.1. Nurse C

*Thoughts of Nurse C*: “Why me? I am a nurse who has worked in my department for over 5 years. I have friends in the department. I am a coordinator of the journal club in the department, and a partner in conducting research. Why do I have to prove myself again in another department?” *Emotions of Nurse C*: Sadness, frustration, fear of failure. *Behavior of Nurse C*: Absence due to illness.

### 5.2. Nurse D

*Thoughts of Nurse D*: “I’m not good enough. I worked hard but, I guess, not hard enough. I need to work harder”. *Emotions of Nurse D*: Stress, worry, lack of self-confidence. *Behavior of Nurse D*: Excessive punctuality, repeating and checking over and over again all professional actions.

*Consequences*: A significant decrease in functioning of the department with an increase in work-related errors.

### 5.3. Case Analysis

Case II describes an example of a process in which the nursing administration of the hospital makes a decision about a department with consequences for the unit nurses and supervisor. The nurses do not know the reasons and motives for this change, receiving a decision “from the higher-ups”. The lack of clarity and transparency in this decision raise various thoughts, emotions, and behaviors in each of the nurses, as described. The difference in the reactions of each of the nurses depends on their core beliefs.

### 5.4. Analysis—Nurse C

Nurse C is afraid of the future. His irrational thoughts are that “other people will take advantage of me and if people really know me, they wouldn’t like me”. His core beliefs will lead him eventually to believe that “no one will love me”. He does not have enough confidence about himself and his ability to prove himself in the future. This contributes to his fear of switching to a new department.

### 5.5. Analysis—Nurse D

Nurse D’s thoughts are that “I’m not good enough. No matter what I do I won’t succeed”. Her core beliefs ultimately are that “I am worthless… and this is why they might take me to another department. I’m not good enough here”. Her core belief is that “I’m nothing. No matter what I do”.

## 6. Use of CB Techniques for Case II

This case reflects two issues, both the response of the supervisor who is being directed from above, and her transferring of this communication to her unit. As a transformational leader, she needs to operationalize the components of transformational leadership. She must take into account her own views and values, and also her staff members, possibly using “modeling” as a way to show leadership, as expressed in the *Idealized Influence* of transformational leadership. From the perspective of the nurse supervisor, dealing with nurses’ negative beliefs requires the nurse supervisor to offer the staff nurses an opportunity to verbalize their feelings about the change, and to be familiar with organizational protocols and needs. This level of transparency could be used as a management tool to improve the approach of the administrators and supervisors in order to plan and carry out decisions made from “top down” smoothly. During the COVID-19 pandemic, for example, nursing management was required to face complex challenges of unit needs for manpower, some of which also existed during routine times [12]. The above case describes a situation in which the behavior of the administration, and their representation by the supervisor, caused maladaptive and unwanted behaviors in Nurses C and D.

Supervisors who choose a non-transparency style may feel pressured to make a decision due to time constraints, or have this style innately part of their personal communication approach. Nurse administrators and supervisors can choose a non-transparent decision-making style when leading their subordinates, dealing with the subsequent outcomes. However, supervisors and administrators should realize that when acting with transparency, when possible, the nursing staff could feel empowered, which may alleviate the anxiety and stress that triggers negative feelings. Making decisions with transparency offers the nurses an understanding of the priorities, necessities, and requirements of a department, possibly over personal desires or preferences, which could contribute to the *Intellectual Stimulation* of transformational leadership.

The case described above presents different options for the supervisor. The supervisor who chooses not to act transparently with the nursing staff regarding the decision of who transfers departments and who does not, may reflect the nurse supervisor’s own anxiety, with concerns for the worst-case scenario of strong staff reactions [43]. By evaluating options for decision-making, and choosing what is best for the employees, all relationships may improve [44]. A nurse supervisor who employs transparency with the nursing staff, sharing with whom and how, and in what manner the change will occur, not only addresses the current issues, but also promotes good decision-making and benefits in the long run. Transparency can improve supervisors’ awareness of relationships among the nursing staff and encourage the development of a more satisfactory work environment [45].

In addition, when looking at the flip side, the supervisor’s beliefs play a role here as well. Personal awareness of values helps when deciding priorities. The supervisor may have a core belief that “by knowing my own values, I become more aware of my choices and more confident with them”. The nurse supervisor’s self-awareness of their values can help direct engagement in elements related to these values, which is important to the nurse supervisor, and can, thus, prevent burnout at this level as well [46].

## 7. Discussion

This paper demonstrates the application of transformational-leadership-like interventions using CB techniques. The “I’s” found in transformational leadership can be operationalized with CB techniques, as illustrated by several real-life cases. These types of cases are found in nursing units world-wide and show how approaching the problem from a humanistic perspective such as that seen with the *Individual Consideration* of the transformational leadership approach, means that the nurses are able to work through their problems and end up with a positive solution. Improving relationships at any level improves the entire environment, leading to a more positive state of functioning.

CB techniques can be aligned with transformational leadership interventions for navigating mentoring and role-modeling. All of these strategies can ultimately be integrated into workshop topics as part of transformational nursing leadership skill-building [40]. Nurse supervisors are working nurses who can provide guidance through processes based on CB techniques. Working together using these techniques has the potential to forge a cooperative relationship based on trust and respect, which, overall, provides a healthy and positive work environment. Depending on the presenting problem and the nurse’s institutional, organizational, and cultural background, there are several aspects of CB techniques that may be adapted [47].

Nurse workplace stress and burnout has been discussed in the literature [48]. Studies have found that the sources of stress and burnout among nurses are mainly both organizational and personal [45]. Previously published interventions for supervisors to consider were reducing workload or increasing autonomy for nurses [49] by empowering a work– life balance [40]. CB techniques, as an application of transformational leadership, promote positive thinking [45], encourage self-compassion, enable problem-solving, and alleviate distress symptoms [50].

*Positive visions and thoughts*, instead of negative ones, are encouraged by CB techniques. Perceiving future situations as only negative can be disruptive [41]. Pondering and focusing on the gloomy picture of the health professional’s future may evoke, especially in young nurse professionals, negative feelings and worries about the future [51]. Supervisors can direct positive thoughts from bad or frightening thoughts noted in both the novice and experienced nurse [52]. Novice nurses have huge potential. They can be encouraged to explore new, improved, relevant values and abilities [53]. Through CB techniques, a positive vision can be conjured concerning the future, without determining an absolute “dangerous world”, as Nurse B presented in her core beliefs, in scenario I.

*Normalization* can be used as a CB technique for the management of errors. Nurses, in their desire to provide excellent care, can be perfectionists. When making an error or a near-error, the nurse can feel anxious about returning to providing patient care, over-think, or be ridden with self-guilt. These feelings can even result in sleep disturbances, further interrupting the effectiveness of the workday and, consequently, presenting with continuous worry about possibly repeating the error. Normalization can be offered by a supervisor by encouraging the realization that errors are made. By normalizing the situation, the nurse may reduce their feelings of guilt and their obsession over the situation. This is a personal as well as a professional intervention that could contribute to the prevention of burnout [54].

Awareness of values helps when advising about personal priorities. By knowing one’s own values, one becomes more aware of one’s choices and feels more comfortable with the choice made. The inherent potential in knowing the values of the nurse can help direct engagement in elements related to these values, which are important to the nurse, thus preventing burnout over time [46]. Awareness of values helps when advising about personal priorities. A potential challenge to this approach is the time and investment required to train the nurse leader to integrate these strategies into their leadership approach. It requires knowledge of the evidence and the strategies, simulated learning, and support to truly immerse the leader into the employment of CB techniques. These CB techniques are to be part of the tool box of the leader to be “pulled out” or “reserved” for when deemed relevant.

One of our main tasks as nursing supervisors in the healthcare system is caring for our nurses. In addition to being an individual, but also a team player, transformational leadership offers *Intellectual Stimulation* as a process for improvement. This includes the ability to filter out negative emotional distress and evaluate the nurse’s cognitive and emotional regulation strategies [55]. Emotional regulation is effective in reducing the occupational stress of nurses [56] and, possibly, can improve job satisfaction [57]. Ultimately, job satisfaction can lead to better care for patients and improved healthcare outcomes. The relationship between nursing job satisfaction and the use of CB techniques as the nurse leader’s approach still need to be examined.

## 8. Conclusions

CB techniques can be used by nursing supervisors as the active process for transformational nursing leadership in several different contexts. Using cognitive behavioral techniques can give the supervisors the ability to help themselves and other nurses as well. As demonstrated, CB techniques can be offered to supervisors as part of their skill set for retention and prevention of burnout for themselves and for their subordinates. It can help manage conflicts and improve communication skills. These techniques can have a significant impact on self-esteem, both professionally and personally in everyday life. By integrating CB techniques into management support programs, the supervisor’s skills will be enriched. These techniques can affect decision-making style, self-esteem, motivation, confidence, and assertiveness. Nurse supervisors can easily employ this framework for better management contributing to an individual’s sense of well-being and job satisfaction [58].

### Implications for Practice/Research

Healthcare organizations should consider incorporating CB techniques into their leadership development programs. This significantly broadens the application of CB techniques from something managers do, to something that is more similar to a change in work culture where these techniques are utilized throughout the organization and are not the responsibility of a few people. Suggested practical ways to implement these techniques in daily nursing management include developing structured training programs with workshops and personal coaching. Also, implementing CB techniques, such as automatic thought identification and problem-solving techniques, help create supportive work environments.

This perspective on CB techniques in nursing leadership contributes to theoretical knowledge in several ways. First, it extends the application of the cognitive behavioral approach beyond its traditional therapeutic context into organizational leadership in healthcare settings. Second, it provides a theoretical basis for understanding how cognitive restructuring and behavioral modification techniques can enhance leadership capabilities in healthcare settings. Ultimately, while the benefits of CB techniques have been clearly elucidated, successfully adding these skills to the supervisors “tool box” will require financial incentives that will make it worthwhile for healthcare organizations to support such programs. This integration of CB techniques into nursing leadership theory offers a new lens by which, through interventional research, the relationship between leadership approaches and organizational outcomes in healthcare may continue to be examined.

## Figures and Tables

**Table 1 nursrep-15-00091-t001:** Cognitive behavioral techniques, strategies, and expected results.

Strategy Type	Cognitive Behavioral Approach	Cognitive Behavioral Technique for Leader–Workplace and Personal Relationships	Expected Result
Cognitive	ABC Model Core beliefs Irrational thoughts	Positive vision	Improved communication skills
		Negative filter	
		Negative view of future	
	Self-Expansion Model	Assertiveness	Increased self-awareness
		Decision-making style	
		Confidence	
Behavioral	Modeling	Modeling	Behavioral change
Emotional	The End Method	Normalization	Increased motivation
	Self-control	Emotional regulation	Stress-coping
	Values	Awareness of personal values Self-compassion	Personal customization priorities

## Data Availability

Data sharing is not applicable. No new data were created or analyzed in this study.

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
