# Peer review of "Creating a Supportive Work Environment: A Cognitive Behavioral Approach for Nurse Leaders"

_nursrep, 2025, doi:10.3390/nursrep15030091_

Round 1
Reviewer 1 Report (Previous Reviewer 3)
Comments and Suggestions for Authors
Dear Authors:
Thank you for your amendments. All the comments were provided for improvements, nothing personal. If you have the freedom to show disagreement with reviewers comments, as your mindset in research and my mindset in research are not the same. I have just one observation, which is regarding your gap or problem statement, as you have analyzed two cases.
”It has been reported that nurse managers lack adequate skills to meet transformational roles (Lyle-Edrosolo et al., 2023).
The above sentence cannot be a gap. The problem already highlighted by Lyle-Edrosolo et al., 2023. The gap should include why the nurse managers lack adequate skills to meet transformational roles. What are those factors/reasons, and how do the nurse managers struggle due to those factors? What is the novelty of your research compared to prior studies?
Thank you, and best wishes.
Author Response
Reviewer 1 |
Comment |
Response |
1 |
I have just one observation, which is regarding your gap or problem statement, as you have analyzed two cases. ”It has been reported that nurse managers lack adequate skills to meet transformational roles (Lyle-Edrosolo et al., 2023). The above sentence cannot be a gap. The problem already highlighted by Lyle-Edrosolo et al., 2023. The gap should include why the nurse managers lack adequate skills to meet transformational roles. What are those factors/reasons, and how do the nurse managers struggle due to those factors? What is the novelty of your research compared to prior studies?
|
Added line 36-39: “They struggle to fulfill transformative roles due to a lack of adequate training and preparation for the role, absence of tailored leadership development programs, at times, a toxic work environment including bullying and poor relationships, and lack of systemic support for their mental and physical health” |

Reviewer 2 Report (New Reviewer)
Comments and Suggestions for Authors
Abstract: Good, succinct and well written.
Stylistic comment regarding Table 1: Be consistent with the use of capital letters throughout the table as this gives a more professional impression.
Line 68-73: Excellent that authors acknowledge that leadership is more than a technical exercise, it also requires an enhancement of what has previously known as, ‘soft skills’, which in itself is a solid argument for the implementation of CB techniques.
Line132-134: Considering this is about managers it is worth mentioning that conflicts can arise between managers and staff, which is quite common. This is of particular importance as there is a power difference between these two parties that can have a negative influence on the success of a CB program. Therefore, a reflection on this would be really valuable in this context.
Line 195-198: The manager may not be in the best position to provide this feedback unless the person works in close proximity to the nurse. It may appear more feasible to allocate this role to the person supervising the nurse in her day-to-day practices.
Throughout the text there is a change between ‘nurse manager’ and ‘nurse supervisor’. It is not clear whether this is the same person or not. This needs to be clarified in the paper.
Line 311-319: In the discussion it appears that CB techniques in the nursing context and training/upskilling/supporting nursing staff can be the responsibility of nurse managers or experienced nurses working with novice nurses. This significantly broadens up the application of CB techniques from something managers do, to something that is more similar to a change in work culture where these CB techniques are utilized throughout the organization and not the responsibility of a few people, where the latter involves a more comprehensive approach. This is not something that the authors reflected upon and what implications this could have on nursing practice.
Line354-359: The previous comment applies to what the authors discuss in this section. Stating that “these CB techniques are to be part of the toolbox of the leader” is well and fine, but what is required for the leader to adopt this? This tends to be the crux of any implementation of something new in organizations, what will it take to convince the leader to invest time, and thereby money, in CB techniques considering that this person is likely already overworked. What are the efficiencies, short- and long-term, that can come out of the implementation of these proposed practices? We in the caring professions tend to highlight, just as the authors have done, that it will reduce stress, improve job satisfaction, lead to better care, and so on. But having spent more than two decades working with organizations in both the public and private sectors, it will come down to the financial incentives, because this is what the health administrators are interested in. It is the lack of converting ‘soft’ goals into financial gains that has been the undoing of many programs. This is something I recommend the authors take into consideration, either in this paper or in future research.
Overall, this is a well-written paper with a lot of merit. Considering the feedback will have the potential to fill the few gaps that has been identified in the authors suggested program.

Author Response
Reviewer 2 |
Comment |
Response |
1
|
Abstract: Good, succinct and well written. |
Thank you |
2 |
Stylistic comment regarding Table 1: Be consistent with the use of capital letters throughout the table as this gives a more professional impression.
|
Corrected p.26 |
3 |
Line 68-73: Excellent that authors acknowledge that leadership is more than a technical exercise, it also requires an enhancement of what has previously known as, ‘soft skills’, which in itself is a solid argument for the implementation of CB techniques.
|
Added line 81: The aim of this paper is to offer unique suggestions for nurse managers, focusing on Cognitive Behavioral techniques that gives an interventional framework for managers promoting a work environment with what was known as ‘soft skills’, that encourages open communication, collaboration, and active management of challenges |
4 |
Line132-134: Considering this is about managers it is worth mentioning that conflicts can arise between managers and staff, which is quite common. This is of particular importance as there is a power difference between these two parties that can have a negative influence on the success of a CB program. Therefore, a reflection on this would be really valuable in this context.
|
Added in line 134-135: “between supervisor and staff nurses”
We have included this group in the listing. The difference in “power” spoken about is not a CB technique issue since these techniques encourage respect and decompression of the stressful situation and focuses the discussion on the issue and not the person. (see the follow up sentences in the text.) |
5 |
Line 195-198: The manager may not be in the best position to provide this feedback unless the person works in close proximity to the nurse. It may appear more feasible to allocate this role to the person supervising the nurse in her day-to-day practices. Throughout the text there is a change between ‘nurse manager’ and ‘nurse supervisor’. It is not clear whether this is the same person or not. This needs to be clarified in the paper.
|
Changed all “manager” to “supervisor” |
6 |
Line 311-319: In the discussion it appears that CB techniques in the nursing context and training/upskilling/supporting nursing staff can be the responsibility of nurse managers or experienced nurses working with novice nurses. This significantly broadens up the application of CB techniques from something managers do, to something that is more similar to a change in work culture where these CB techniques are utilized throughout the organization and not the responsibility of a few people, where the latter involves a more comprehensive approach. This is not something that the authors reflected upon and what implications this could have on nursing practice.
|
Line 313- the referral to experienced nurses also using CB techniques has been removed.
Added in line 380 – 384: ”This significantly broadens up the application of CB techniques from something managers do, to something that is more similar to a change in work culture where these techniques are utilized throughout the organization and not the responsibility of a few people”. |
7 |
Line 354-359: The previous comment applies to what the authors discuss in this section. Stating that “these CB techniques are to be part of the toolbox of the leader” is well and fine, but what is required for the leader to adopt this? This tends to be the crux of any implementation of something new in organizations, what will it take to convince the leader to invest time, and thereby money, in CB techniques considering that this person is likely already overworked. What are the efficiencies, short- and long-term, that can come out of the implementation of these proposed practices? We in the caring professions tend to highlight, just as the authors have done, that it will reduce stress, improve job satisfaction, lead to better care, and so on. But having spent more than two decades working with organizations in both the public and private sectors, it will come down to the financial incentives, because this is what the health administrators are interested in. It is the lack of converting ‘soft’ goals into financial gains that has been the undoing of many programs. This is something I recommend the authors take into consideration, either in this paper or in future research.
|
Thank you. Added in line 394 – 399: “Ultimately, while the benefits of CB techniques have been clearly elucidated, successful implementation of adding these skills to the supervisors “tool box” will require financial incentives that will make it worthwhile for healthcare organizations to support such programs. This integration of CB techniques into nursing leadership theory offers a new lens by which through interventional research, the relationship between leadership approaches and organizational outcomes in healthcare may continue to be examined.”
|

Reviewer 3 Report (New Reviewer)
Comments and Suggestions for Authors
nursrep-3399406-peer-review-v1
The purpose of reviewed paper is to “identifying cognitive behavioural (CB) techniques that can help nurse managers more effectively navigate interpersonal challenges, reduce workplace stress, improve team cohesion, and ultimately enhance overall organizational performance and staff well-being” (line numbers: 8-10). The paper contains a lot of weaknesses such as: literature review, research methods, contrived practical examples, text structure, comprehensibility of statements.
Originality/Novelty
One of the keywords of the peer-reviewed scientific paper is “Transformational Leadership”. The paradigm of transformational leadership emerged in leadership theory half a century ago – so it is not new knowledge and is only partly up-to-date, as the new paradigm includes and transcends previously existing paradigms. During the evolution of civilisation, leadership paradigms have emerged, such as:
· from antiquity to the 1970s. – classical leadership and charismatic leadership,
· from 1970s to mid-1980s. – transactional leadership and transformational leadership,
· since 1970 – servant leadership based on ethical leadership,
· from the mid-1980s to 2000. – visionary leadership,
· since 2000 – organic leadership, distributed leadership based on self-leadership (leading oneself),
· since 2003 – authentic leadership,
· since 2004 – sustainable leadership.
The research question was not defined in the paper and the connections between the transformational leadership paradigm and the cognitive behavioural techniques were not shown. Furthermore, cognitive behavioural techniques do not have a definition.
Significance
Low.
Quality of Presentation
The scientific paper was not written in an appropriate manner, the literature review was not adequately presented and the paper does not meet the highest standards of presentation of the results used. The following elements of the text should be reconsidered:
1) the paper lacks a section describing the research methods. There is only information in the abstract that indicates that the 2 cases described in the empirical part were invented by the authors: “Approach: Through a comprehensive review of leadership literature and clinical management practices, CB techniques can be integrated into nursing management.
Two hypothetical scenarios within context are offered where CB techniques can enhance nursing leadership effectiveness”,
2) the words contained in the topic of the paper should be defined in the body of the paper, e.g. empowerment,
3) the purpose of reviewed paper is to “identifying cognitive behavioural (CB) techniques that can help nurse managers more effectively navigate interpersonal challenges, reduce workplace stress, improve team cohesion, and ultimately enhance overall organizational performance and staff well-being” (line numbers: 8-10). In line numbers 77-80, the authors wrote (quote): “The aim of this paper is to offer unique suggestions for nurse managers, focusing on Cognitive Behavioral techniques that gives an interventional framework for managers promoting a work environment that encourages open communication, collaboration, and active management of challenges”. This (“offer unique suggestions”) cannot be the purpose of scientific work. The purpose of the paper should be elaborated in the introduction and the research questions that are missing should be identified,
4) don’t write in “1. Introduction” about the ABC model if you only explain it in “2. What are CB techniques?”,
5) the “6. Discussion” section of the text contains many valuable and up-to-date research findings. However, there is a lack of a solid literature review at the beginning
of the research paper, in which the choice of research topic is motivated,
6) in line number 56, the authors wrote (quote): “In an organic interface with the Cognitive Behavioral Model (Table 1)”, but there is no model in Table 1. in line number 64, the authors wrote (quote): “Table 1. CB Techniques as nursing tools” but the subject of the paper is not “nursing tools” but “Empowering Nurse Leadership” – nursing management. What is included in Table 1.? This can only be learned from the fourth page of the text: “Table 1 presents a summary of CB techniques and their expected outcomes that can be suggested for nursing managers” (line numbers 129-130). It is too chaotic a text structure for the viewer and makes it difficult to understand. The methods indicated in Table 1. should be defined directly below and the sources of these definitions should be given. Describing the methods only in the research section (Case I and Case II), is not good practice. Moreover, the source of Table 1. should be indicated,
7) bibliography items 39-40 to be corrected in line numbers: 487-489: „39. Rose, V., & Perz, J. (2005). Is CBT useful in vocational rehabilitation for people with a psychiatric disability? Psychiatric Rehabilitation 487 Journal, 29(1), 56. 40. https://doi.org/10.2975/29.2005.56.58 489”,
8) bibliography items 39-40 to be corrected in line numbers: 487-489:
„39. Rose, V., & Perz, J. (2005). Is CBT useful in vocational rehabilitation for people with a psychiatric disability? Psychiatric Rehabilitation 487 Journal, 29(1), 56.
40. https://doi.org/10.2975/29.2005.56.58 489”.
9) the formatting of parts of the bibliography needs improvement (line numbers: 427-431):
“12. Field, T. A., Beeson, E. T., & Jones, L. K. (2015). The new ABCs: A practitioner's guide to 427
13. neuroscience-informed cognitive-behavior therapy. Journal of Mental Health
14. Counseling, 37(3), 206-220. http:// doi:10.17744.mehc.37.3.02
15. Fischer, S. A. (2016). Transformational leadership in nursing: a concept analysis. Journal of
a. advanced nursing, 72(11), 2644-2653. https://doi.org/10.1111/jan.13049”.
The lack of the following sections in the text: literature review and research methods
as well as invented practical examples are the reviewer’s main criticisms, which resulted
in a negative assessment of the entire scientific work.
Scientific Soundness
Low.
Interest to the Readers
Low.
Overall Merit
Low.
English Level
The English language is appropriate and understandable. However, there are minor shortcomings language in the paper:
· the summary of the work lacks full stops at the end of sentences, e.g.: „…well-being Approach:…” (in line number 11) and “healthcare settings” (in line number 24),
· “work- life balance” instead of “work- life balance” (line number 325),
· text formatting.

Comments on the Quality of English Language
Accept after minor revisions (corrections to minor methodological errors and text editing).
Author Response
Reviewer 3 |
Comments |
Response |
1 |
Originality/Novelty One of the keywords of the peer-reviewed scientific paper is “Transformational Leadership”. The paradigm of transformational leadership emerged in leadership theory half a century ago – so it is not new knowledge and is only partly up-to-date, as the new paradigm includes and transcends previously existing paradigms.
The research question was not defined in the paper and the connections between the transformational leadership paradigm and the cognitive behavioural techniques were not shown. Furthermore, cognitive behavioural techniques do not have a definition.
|
1. Transformational leadership can still be found in the present-day literature Examples: · Collins, E., Owen, P., Digan, J., & Dunn, F. (2020). Applying transformational leadership in nursing practice. Nursing Standard, 35(5), 59-66) · Broome, M.E. (2024). Transformational leadership in nursing: From expert clinician to influential leader. springer publishing company. · Xie, Y., Gu, D., Liang, C., Zhao, S., & Ma, Y. (2020). How transformational leadership and clan culture influence nursing staff's willingness to stay. Journal of nursing management, 28(7), 1515-1524. 2. There is no research question- this is a theoretical piece as has been explained in previous communication: . 3. CB techniques were defined see Table 1
|
2 |
The scientific paper was not written in an appropriate manner, the literature review was not adequately presented and the paper does not meet the highest standards of presentation of the results used. The following elements of the text should be reconsidered: 1) …the paper lacks a section describing the research methods. There is only information in the abstract that indicates that the 2 cases described in the empirical part were invented by the authors: “Approach:
TH Through a comprehensive review of leadership literature and clinical management practices, CB techniques can be integrated into nursing management.
|
This manuscript is a theoretical piece offering a strategy for transformation leadership skills, using CB techniques, with practical applications to generated cases based in reality, though not driven by something that happened in situ.
Statistical analyses or applied findings are not relevant due to the fact that this manuscript is not a research piece though the background is evidence based and therefore employs a scientific approach
A comprehensive review of the leadership techniques offered in the literature would be useful if that was the focus of this article. Since the focus is on the idea of integrating CB techniques, the review of the literature focuses on that body of knowledge. Leadership transformation literature is referred to in the text.
|
3 |
the words contained in the topic of the paper should be defined in the body of the paper, e.g. empowerment, |
Title changed to: Creating a Supportive Work Environment: A Cognitive Behavioral Approach for Nurse Leaders
Line 325 – word replacement: “problem solving empowerment” with “enabling problem solving “ |
4 |
the purpose of reviewed paper is to “identifying cognitive behavioural (CB) techniques that can help nurse managers more effectively navigate interpersonal challenges, reduce workplace stress, improve team cohesion, and ultimately enhance overall organizational performance and staff well-being” (line numbers: 8-10). In line numbers 77-80, the authors wrote (quote): “The aim of this paper is to offer unique suggestions for nurse managers, focusing on Cognitive Behavioral techniques that gives an interventional framework for managers promoting a work environment that encourages open communication, collaboration, and active management of challenges”. This (“offer unique suggestions”) cannot be the purpose of scientific work. The purpose of the paper should be elaborated in the introduction and the research questions that are missing should be identified, |
This manuscript is a theoretical piece offering a strategy for transformation leadership skills, using CB techniques, with practical applications therefore if subheadings are used, they would need to represent this manuscript which is not a research based one.
|
5 |
don’t write in “1. Introduction” about the ABC model if you only explain it in “2. What are CB techniques?”, |
The ABC model is mentioned and explained in the introduction chapter and an explanation is immediately provided on pages 4-5 lines 94-101 |
6 |
Discussion” section of the text contains many valuable and up-to-date research findings. However, there is a lack of a solid literature review at the beginning of the research paper, in which the choice of research topic is motivated, |
This manuscript’s significance is in that it is a theoretical piece offering a strategy for transformation leadership skills, using CB techniques, with practical applications. See page 4 line 79-83: “The aim of this paper is to offer unique suggestions for nurse supervisors, focusing on Cognitive Behavioral techniques that gives an interventional framework for supervisors promoting a work environment with what was known as ‘soft skills’, that encourages open communication, collaboration, and active management of challenges.”
Therefore, the introduction concentrates on the model’s background. The discussion section offers evidence to the suggested model after the article has displayed the use of the model. |
7 |
in line number 56, the authors wrote (quote): “In an organic interface with the Cognitive Behavioral Model (Table 1)”, but there is no model in Table 1. in line number 64, the authors wrote (quote): “Table 1. CB Techniques as nursing tools” but the subject of the paper is not “nursing tools” but “Empowering Nurse Leadership” – nursing management. |
Cognitive Behavioural Model (Table 1) title changed to Cognitive Behavioural Approach (line 60 and table 1).
Title has been changed to reflect content |
8 |
The methods indicated in Table 1. should be defined directly below and the sources of these definitions should be given. Describing the methods only in the research section (Case I and Case II), is not good practice. Moreover, the source of Table 1. should be indicated |
The guidelines for placing the table are provided by the journal, usually at the end of the text.
The table is offered as a visual organization of what is described in the text. The text offers the references for the concepts described in the table.
|
9 |
Re Refer to table and its content in the introduction |
See line 131 in the introduction |
10 |
7) bibliography items 39-40 to be corrected in line numbers: 487-489: „39. Rose, V., & Perz, J. (2005). Is CBT useful in vocational rehabilitation for people with a psychiatric disability? Psychiatric Rehabilitation 487 Journal,29(1),56. https://doi.org/10.2975/29.2005.56.58 489”,
|
Thank you. Corrected line 530. |
11 |
the formatting of parts of the bibliography needs improvement (line numbers: 427-431): 12. Field, T. A., Beeson, E. T., & Jones, L. K. (2015). The new ABCs: A practitioner's guide to 427 neuroscience-informed cognitive-behavior therapy. Journal of Mental Health Counseling, 37(3), 206-220. http:// doi:10.17744.mehc.37.3.02 Fischer, S. A. (2016). Transformational leadership in nursing: a concept analysis. Journal of a. advanced nursing, 72(11), 2644-2653. https://doi.org/10.1111/jan.13049”. |
Thank you. Corrected. |

Round 2
Reviewer 3 Report (New Reviewer)
Comments and Suggestions for Authors
The purpose of reviewed paper is to “cognitive behavioural (CB) techniques that 8 can help nurse supervisors more effectively navigate interpersonal challenges, reduce workplace stress, improve team cohesion, and ultimately enhance overall organizational performance and staff well-being” (line numbers: 8-10). The paper contains a lot
of weaknesses such as: literature review, research methods, contrived practical examples and text structure.
Originality/Novelty
One of the keywords of the peer-reviewed scientific paper is “Transformational Leadership”. The paradigm of transformational leadership emerged in leadership theory half
a century ago – so it is not new knowledge and is only partly up-to-date, as the new paradigm includes and transcends previously existing paradigms. During the evolution of civilisation, leadership paradigms have emerged, such as:
- since antiquity – classical leadership and charismatic leadership,
- since 1970 – transactional leadership and transformational leadership,
- since 1970 – servant leadership based on ethical leadership,
- since the mid-1980s. – visionary leadership,
- since 2000 – organic leadership, distributed leadership based on self-leadership (leading oneself),
- since 2003 – authentic leadership,
- since 2004 – sustainable leadership.
The research question was not defined in the paper and the connections between
the transformational leadership paradigm and the cognitive behavioural techniques were not shown. Furthermore, cognitive behavioural techniques do not have a definition.
The biocratic leadership paradigm currently discussed in the literature fits the publication (Bloom, S. L. (2023). A Biocratic Paradigm: Exploring the Complexity of Trauma-Informed Leadership and Creating PresenceTM. Behavioral Sciences, 13(5), 355, 1–22. https://doi.org/10.3390/bs13050355).
Significance
Low.
Quality of Presentation
The scientific paper was not written in an appropriate manner, the literature review was not adequately presented and the paper does not meet the highest standards of presentation
of the results used. The following elements of the text should be reconsidered:
- the paper lacks a section describing the research methods. There is only information in the abstract that indicates that the 2 cases described in the empirical part were invented by the authors: “Approach: Through a comprehensive review of leadership literature and clinical management practices, CB techniques can be integrated into nursing management.
Two hypothetical scenarios within context are offered where CB techniques can enhance nursing leadership effectiveness”. Invented examples of nurses’ problems can be successfully used in training material. In a scientific article, on the other hand, it would be desirable to rely on real-life problems, descriptions of which can be obtained by the interview method. The authors did not take the trouble to talk to the nurses about the real dilemmas they experienced during their professional work. If the researchers failed to do this, how are the nurse supervisors supposed to succeed? - the text lacks research questions,
- the “6. Discussion” section of the text contains many valuable and up-to-date research findings. However, there is a lack of a solid literature review at the beginning
of the research paper, in which the choice of research topic is motivated, - the source of Table 1. should be indicated,
- bibliography items 38-39 to be corrected in line numbers: 500-502: „38. Rose, V., & Perz, J. (2005). Is CBT useful in vocational rehabilitation for people with a psychiatric disability? Psychiatric Rehabilitation 487 Journal, 29(1), 56. 39. https://doi.org/10.2975/29.2005.56.58 489”,
- the formatting of parts of the bibliography needs improvement (line numbers: 441-444): The lack of the following sections in the text: literature review and research methods as well as invented practical examples are the reviewer’s main criticisms, which resulted in a low assessment of the entire scientific work.
A scientific paper based on a literature review requires a description of the research method, e.g. a systematic review. The idea is that researchers who read an article will be able to perform analogous research if they explore the same topic. To this end, research questions are also formulated. This was not done by the Authors of the reviewed text.
Scientific Soundness
Low.
Interest to the Readers
Low.
Overall Merit
Low.
English Level
The English language is appropriate and understandable. However, there are minor shortcomings language in the paper:
- the summary of the work lacks full stops at the end of sentences, e.g.: „…well-being Approach:…” (in line number 11) and “healthcare settings” (in line number 24),
- text formatting.

Author Response
We appreciate the reviewer's desire to improve our paper.
There seems to however be a misunderstanding with the reviewer as to the nature of our paper, issues that we responded to in our response to comments that were already submitted.
We would like to reiterate certain important clarifications:
- "The paradigm of transformational leadership emerged in leadership theory half a century ago so it is not new knowledge and is only partly up-to-date, as the new paradigm includes and transcends previously existing paradigms."
Response:
Transformational leadership is a term still found and discussed in today's literature. It meets synergistically with our concepts. In 2025 alone, and we are only in February, there were over 2000 articles discussing this concept. (See Google Scholar). Therefore, we do not see this as an "out of date" concept.
- "The research question was not defined in the paper"
".... the paper does not meet the highest standards of presentation of the results used. The following elements of the text should be reconsidered: 1) the paper lacks a section describing the research methods..."
"...the text lacks research questions"
" the paper does not meet the highest standards of presentation of the results"
Response:
This manuscript is a theoretical piece offering a strategy for transformation leadership skills, using CB techniques, with practical applications to generated cases based in reality, though not driven by something that happened in situ. A research based methods section with statistical analyses or applied findings is not relevant due to the fact that this manuscript is not a research piece though the background is evidence based and therefore employs a scientific approach
- "... the connections between the transformational leadership paradigm and the cognitive behavioural techniques were not shown."
Response:
Leadership transformation literature is referred to in the text as a trigger for discovering new techniques for leadership strategies. CB techniques is the suggested strategy in this article. This is not a review piece of transformational leadership.
- "Furthermore, cognitive behavioural techniques do not have a definition."
Response: This was clarified in our responses to the reviewers in that the definition of these techniques is offered in Table 1.
*******
We are very interested in publishing in your esteemed journal.
Please advise as to how to move this process forward to meet the needs of the journal while representing, authentically, our manuscript.
This manuscript is a resubmission of an earlier submission. The following is a list of the peer review reports and author responses from that submission.
Round 1
Reviewer 1 Report
Comments and Suggestions for Authors
The topic of the paper is relevant and interesting, but it lacks scientific approach, hence it cannot be accepted for publication.
In this paper, the author relies on two case studies to illustrate transformational leadership and cognitive-behavioural techniques in health care settings. These examples are from specific nursing environments, and the author has not explained how they are representative of the broader field, has not discussed how these cases reflect the wider nursing profession or how different cultural, institutional, or organizational variables could influence the outcomes. Therefore, this approach lacks broader empirical validation and cannot produce generalized findings or definitive conclusions.
Apart from the lack of a clear methodology that explains how these cases were selected and analysed, there is no statistical evidence or quantitative evaluation of how effective cognitive-behavioural techniques are in this settings. This lack of crucial scientific elements undermines the credibility of the conclusions the author has drawn from these cases.
The case studies in this paper do not meet the requirements for the necessary methodological and scientific rigor—they are better suited for a professional or practical discussion in nursing leadership. I therefore recommend rejecting the paper based on these significant shortcomings.
Comments on the Quality of English Language
The English is understandable.
Reviewer 2 Report
Comments and Suggestions for Authors
Thank you for inviting me to review the manuscript titled "Cognitive Behavioral Techniques in Transformational Leadership: Practical Application Using Nursing Case Scenarios." This paper is timely and relevant, offering a novel integration of cognitive behavioral techniques with transformational leadership, aimed at addressing pressing issues such as stress and burnout in nursing. The theoretical underpinnings are robust, and the manuscript benefits greatly from the illustrative case scenarios that enhance its practical value. Despite its strengths, the manuscript presents several areas that could be enhanced to improve its clarity and impact:
- Purpose and Rationale: The introduction does not clearly articulate why it is critical to study cognitive behavioral techniques within the context of transformational leadership. A clearer justification for the study, outlining both the academic and practical implications, would strengthen the manuscript.
- Connection to Burnout and Stress: The sudden mention of burnout and stress in lines 51-59 seems somewhat disconnected from the preceding content. Clarifying how these elements are directly related to the central theme of the manuscript would help in maintaining coherence and reinforcing the narrative.
- Selection and Representation of Case Studies: The rationale behind the selection of the two case studies presented is not clearly stated. Expanding on why these particular cases were chosen, their representativeness, and whether they are based on real events or are hypothetical would add depth and relevance to the discussion.
Furthermore, methodological enhancements could also be considered:
- Detailed Methodology: The manuscript would benefit from a more explicit explanation of how the cognitive behavioral techniques are applied in leadership practices within real-world nursing settings.
- Empirical Support: Incorporating empirical evidence or outcomes from real case studies where these techniques have been applied could substantiate the claims and strengthen the practical recommendations.
- Discussion on Limitations: Expanding the discussion to include potential challenges, limitations of the approach, and future research directions would provide a more balanced view and enrich the manuscript.
In conclusion, while the manuscript provides valuable insights into the integration of cognitive behavioral techniques in nursing leadership, addressing these areas will significantly enhance its contribution to the field.
Comments on the Quality of English Language
Thank you for inviting me to review the manuscript titled "Cognitive Behavioral Techniques in Transformational Leadership: Practical Application Using Nursing Case Scenarios." This paper is timely and relevant, offering a novel integration of cognitive behavioral techniques with transformational leadership, aimed at addressing pressing issues such as stress and burnout in nursing. The theoretical underpinnings are robust, and the manuscript benefits greatly from the illustrative case scenarios that enhance its practical value. Despite its strengths, the manuscript presents several areas that could be enhanced to improve its clarity and impact:
- Purpose and Rationale: The introduction does not clearly articulate why it is critical to study cognitive behavioral techniques within the context of transformational leadership. A clearer justification for the study, outlining both the academic and practical implications, would strengthen the manuscript.
- Connection to Burnout and Stress: The sudden mention of burnout and stress in lines 51-59 seems somewhat disconnected from the preceding content. Clarifying how these elements are directly related to the central theme of the manuscript would help in maintaining coherence and reinforcing the narrative.
- Selection and Representation of Case Studies: The rationale behind the selection of the two case studies presented is not clearly stated. Expanding on why these particular cases were chosen, their representativeness, and whether they are based on real events or are hypothetical would add depth and relevance to the discussion.
Furthermore, methodological enhancements could also be considered:
- Detailed Methodology: The manuscript would benefit from a more explicit explanation of how the cognitive behavioral techniques are applied in leadership practices within real-world nursing settings.
- Empirical Support: Incorporating empirical evidence or outcomes from real case studies where these techniques have been applied could substantiate the claims and strengthen the practical recommendations.
Discussion on Limitations: Expanding the discussion to include potential challenges, limitations of the approach, and future research directions would provide a more balanced view and enrich the manuscript.
In conclusion, while the manuscript provides valuable insights into the integration of cognitive behavioral techniques in nursing leadership, addressing these areas will significantly enhance its contribution to the field.
Reviewer 3 Report
Comments and Suggestions for Authors
Comments:
1) An abstract must contain the research purpose, methods, findings, and recommendations. Thus, authors are requested to rewrite the abstract as per the guideline. The present abstract is unclear.
2) In the introduction section, the authors must discuss the research gaps, the consequences of cognitive behavioral techniques in transformational leadership in the nursing sector, the need for cognitive behavioral techniques, and the significance of the research. The present research overlooks all these issues.
3) Besides, this paper aims to offer unique suggestions for nurse managers, focusing on cognitive behavioral techniques that provide an interventional framework for managers, promoting a work environment that encourages open communication, collaboration, and active management of challenges. Unfortunately, the purpose of this research is not aligned with the research title, which is “Cognitive behavioral techniques in transformational leadership: Practical application using nursing case scenarios”
4) A comprehensive literature review, beyond just 'What are CB techniques?', is crucial. It provides the theoretical framework for the study and explains the necessity and application of cognitive behavioral techniques in transformational leadership in the nursing sector.
5) This study completely overlooked the methodology section, which is one significant drawback. Research methodology is mandatory, whether it is a qualitative, quantitative, or case-based study. Presently, the authors provided 2 cases of nurses:
(i) Case I: Management within the department
(ii) Case II: Decision-making for senior nursing management
What are the reasons for choosing these two cases, as you did not discuss these concerns under the research gap?
6) This study did not offer any theoretical and practical implications for managers, which are crucial for any research based on their findings. Therefore, it's important to ensure that the study provides both theoretical and practical implications for managers, as these are crucial for the application of the research findings in real-world scenarios.
7) This study also did not propose limitations and scope for further research. Thus, the authors are requested to add the points mentioned above.
Comments on the Quality of English Language
Moderate editing of the English language is required.